# Implementing a Community-Based Initiative to Improve Nutritional Intake among Home-Delivered Meal Recipients

**DOI:** 10.3390/nu14050944

**Published:** 2022-02-23

**Authors:** Lisa A. Juckett, Govind Hariharan, Dimitri Camargo Dodonova, Jared Klaus, Melinda Rowe, Elana Burak, Benetta Mason, Leah Bunck

**Affiliations:** 1School of Health and Rehabilitation Sciences, The Ohio State University, Columbus, OH 43210-2205, USA; 2Coles College of Business, Kennesaw State University, Kennesaw, GA 30144-0405, USA; gharihar@kennesaw.edu (G.H.); dcamargo@kennesaw.edu (D.C.D.); 3Lifecare Alliance, Columbus, OH 43223-1809, USA; jklaus@lifecarealliance.org (J.K.); mrowe@lifecarealliance.org (M.R.); eburak@lifecarealliance.org (E.B.); bmason@lifecarealliance.org (B.M.); lbunck@lifecarealliance.org (L.B.)

**Keywords:** malnutrition, homebound, home- and community-based services, long-term services and supports, registered dietitian nutritionists

## Abstract

Home-delivered meal (HDM) recipients are a highly vulnerable group of older adults at risk for malnutrition and subsequent health decline. To help HDM recipients increase their nutritional intake, HDM agencies may provide expanded meal options that allow older adults to have greater autonomy over their meal selection; however, the extent to which recipients are able to select nutritious meals that are responsive to their health complexities is unknown. This study examined the nutritional content of meals selected by HDM recipients enrolled in an expanded menu plan through a large HDM agency. Data were drawn from a retrospective chart review of 130 HDM recipients who had the option of selecting their own HDM meals and frequency of meal delivery. Findings indicate that older adults who selected their own meals chose meals that were significantly lower in protein, potassium, fat, and calories. The lack of these nutrients suggests that older adults enrolled in expanded menu plans should be referred to registered dietitian nutritionists who can provide skilled guidance in meal selection. To address this need, we also describe and provide preliminary data representing a referral program designed to connect HDM recipients to dietetic services with the goal of optimizing older adult nutrition and health-related outcomes.

## 1. Introduction

Home-delivered meal (HDM) programs aim to provide older adults with nutritional support, particularly those who are food-insecure and at high risk for health decline and hospitalization [1,2]. Across the United States, HDM programs have been effective in improving the dietary quality of older adults, age 60 and over, and contribute to enhanced social engagement, improved wellbeing, and lower rates of nursing home placement [3,4,5]. Recipients of HDMs typically live alone, have a household income of less than 20,000 USD per year, and have a high prevalence of comorbid health conditions, several of which are related to nutrition and dietary behaviors [6,7,8]. For instance, data from the 2019 National Survey of Older Americans Act Participants indicated that 38% of HDM recipients have a self-reported diagnosis of diabetes, 51% have high cholesterol, and 76% are living with hypertension [7]. Without adequate nutritional intake, older adults with these combined health conditions are at substantial risk for additional health complications, such as heart disease, kidney damage, stroke, and diabetic retinopathy [9,10,11].

Despite the critical importance of nutritional intake among high-risk older adult populations, older adults often lack awareness of how to routinely meet their nutritional needs given their health complexities [12,13]. Among HDM recipients specifically, the meal consumption of as many as 80% of older adults was insufficient for meeting recommended energy and macronutrient intake, further elevating the risk of malnourishment [14]. One potential option to improve the nutritional intake of HDM recipients is to expand meal options and provide older adults with greater choice and control over selecting their daily meals [15]. This option aligns with prior research which found that autonomy over food selection and the opportunity to select meals from a variety of food choices are associated with increased food intake in older adult populations [16,17]. Although expanded HDM meal options may hold promise for improving the older adults’ feeling of control over meal selection, the extent to which older adults will be able to make appropriate meal selections given their health complexities is unclear. Certainly, the concepts of malnutrition, dietary quality, and macronutrient intake are quite complex, warranting the need for skilled professionals, i.e., registered dietitian nutritionists (RDNs), who can provide person-centered, dietetic services to improve healthy lifestyle behaviors of older adults in HDM settings [18,19].

While RDNs can provide valuable nutrition-related services to the HDM population, there is widespread variability in the extent to which HDM recipients access care provided by RDNs in the community. For instance, individual State Units on Aging set their own expectations for the type and frequency of nutrition education and counseling that must be provided to HDM recipients. Some states require that nutrition counseling be provided on an as-needed basis by RDNs specifically, while other states provide nutrition education on a monthly basis by non-RDN professionals who have expertise in nutritional support [1,20]. Although nutrition education has been found to improve outcomes for older adults and is an expectation of federally funded nutrition programs [21], in 2019, only 7.6% of HDM recipients indicated that they had a nutrition counselor who offered nutritional advice on the basis of their health conditions [7]. This alarmingly low rate of nutrition counseling provided to HDM recipients is particularly concerning for several reasons. First, it calls into question the extent to which HDM programs are fully providing the nutritional support services mandated by the Older Americans Act—the federal legislation that sets standard expectations for HDM programs. Next, without nutritional counseling and support, HDM programs may be unaware of the degree to which meals are helping older adults in meeting their nutritional needs. Lastly, the health status of HDM recipients is often complex, necessitating skilled dietetic guidance to address older adults’ risk factors for malnourishment, *particularly* among HDM recipients who have the autonomy to select daily or weekly meal items available on expanded meal plans. With these concerns in mind and with the rapid growth of the aging population [22,23], HDM researchers and providers must identify older adults with the highest need for RDN services and establish effective approaches for connecting HDM recipients to RDNs in order to optimize the nutritional status of the HDM community.

To understand how best to integrate RDN services in HDM settings, the present study aims to answer the following research questions: (1) To what extent are HDM recipients able to select nutritious meal options from an expanded HDM meal plan? (2) What are the health characteristics of HDM recipients with the highest need for RDN services? To conclude, we present a referral model, currently being implemented by one HDM agency to streamline RDN services to HDM recipients—a subgroup of low-income older adults at high risk for malnutrition and associated health disparities given their difficulty in routinely accessing specialty healthcare. RDNs could arguably provide HDM recipients with more equitable nutrition services, thereby preventing or delaying malnutrition-related health decline and subsequent institutionalization.

## 2. Materials and Methods

### 2.1. Setting

The partner agency for this study was a nonprofit community-based organization whose nutritional services reach approximately 8000 older adults in the greater Columbus, Ohio area. The agency employs over 200 full- and part-time staff members, including five RDNs who provide nutrition education and counseling to nutrition program participants. The HDM program is embedded within the larger organization that provides additional home- and community-based services including congregate dining, home repairs, wellness screenings, and personal care assistance.

### 2.2. Design and Participants

To answer the two primary research questions, data were drawn from a convenience sample of 248 HDM recipients through a retrospective chart review. Recipients’ health records were eligible for inclusion in the chart review if recipients were over the age of 60 years and had contacted the partner agency’s internal case manager about opting into an expanded menu plan in calendar year 2021. In order to sufficiently answer the research questions, only those HDM recipients with complete health and demographic information were included for analysis.

### 2.3. Expanded Meal Plan

All older adults enrolled in the HDM program at the partner agency initially received daily *hot* or *chilled* meal deliveries. These meals were predetermined by the agency and rotated on a monthly schedule. Recipients who enrolled in the expanded meal plan could receive the default selection of *frozen* meals that were delivered in quantities of five, seven, 10, or 14 meals. At the preference of the expanded meal plan recipient, recipients could—rather than receive the default frozen meals—choose to select their own meals from over 40 meal options that were classified as follows: gluten-free, low-cholesterol, heart-healthy, low-sodium, kosher, and vegetarian. Meals were also available in alternative textures (e.g., mechanical soft) for recipients with dental concerns or swallowing disorders.

### 2.4. Data Sources

Socio-demographics were collected from all HDM recipients as part of their standard evaluation completed upon HDM enrollment. The following socio-demographics were extracted from each recipient’s health record: gender, age, marital status, household composition, and income. Meal selections and recipients’ self-reported health conditions were also collected as were three state-mandated measures used to assess recipients’ functional and nutritional status. Functional status was measured through the Activity of Daily Living (ADL) scale [24] and the Instrumental Activity of Daily Living (IADL) scale [25]—both of which were adapted for use by HDM agencies in the state of Ohio. The adapted versions of the ADL and IADL scales were not validated through psychometric evaluation but rather are used for clinical purposes at the partner agency and must be administered to all new HDM recipients as mandated by the State Unit on Aging. Maximum scores on the ADL scale and IADL scale were 30 points and 33 points, respectively, with higher scores indicating greater levels of functional impairment. Nutrition risk assessment (NRA) was completed using the Disease, Eating poorly, Tooth loss, Economic hardship, Reduced social contact, Multiple medicines, Involuntary weight loss/gain, Needs assistance in self-care, and Elder years above age 80 (DETERMINE) Checklist [26] measured nutritional risk among participants. DETERMINE Checklist scores of 0–2 points = good nutritional health, 3–5 points = moderate nutritional health, and 6+ points = high nutritional risk. The lead RDN (JK) also developed a custom Excel file containing the nutritional content of each meal item provided by the agency to HDM recipients. Nutritional content consisted of the total values of sodium, protein, fat, cholesterol, potassium, and calories in each meal item (see Appendix A for values).

### 2.5. Calculation of Nutrition Content of Meals

Protein, sodium (Na), fat, cholesterol (CHO), calories, and potassium were calculated for the standard hot/chilled daily meals, for the default frozen meals on the expanded meal plan, and for the frozen meals selected by recipients on the expanded meal plan. To calculate the average nutrition levels for each meal, we averaged the nutritional content levels for each menu item from the menu for the month preceding (during which the recipient received standard hot or chilled meals) the expanded meal plan choice compared to the most recent month they received the expanded meal plan options. The nutritional content levels for recipients who did not opt in to the expanded meal plan but continued to receive the standard hot or chilled meals, were calculated on the basis of the month after they contacted the case manager to inquire about the expanded meal plan option. Other important measures of nutritional content such as iron, fiber, and carbohydrate values were unfortunately not available in the data and represent a limitation. In order to understand the difference in nutritional content being delivered to HDM recipients, we split the sample into those who did not opt in to the expanded meal plan (expanded plan = 0) and those who selected meals from the expanded plan (expanded plan = 1).

### 2.6. Analysis

IBM SPSS Statistics version 28 was used for the statistical analysis. *t*-Tests for independent samples were employed to compare differences of numerical variables between the groups expanded plan = 0 and expanded plan = 1. When there were more than two groups, one-way ANOVA was conducted, with post hoc options Tukey’s-b for equal variances assumed and Games–Howell for equal variances not assumed. A chi-square test was conducted to analyze relationships between two categorical variables, and adjusted standardized residuals were used to identify two-category relationships. A *t*-test of correlation coefficient was used to assess the significance between two numerical variables.

## 3. Results

The initial sample had 248 individuals, from which 17 individuals who were less than 60 years old were removed and 101 individuals were removed for missing age or gender. Of the 101 removed, 100 were for missing both age and gender and one for missing gender alone, resulting in a full sample of 130 individuals to be used in this study. Those with missing age information also had missing information on meals (except for one individual) and, hence, could not be studied to see if there were any significant differences between them and the rest of the sample. Given the small size of the remaining sample with information on all relevant variables, we kept as many as had complete records for each specific aspect of this study; hence, the number of recipients varies across tables and variables of interest. For example, monthly income information was only available for 68 individuals. In addition, five individuals had missing information on their standard (hot/chilled) meals, reducing the sample for the standard plan to 125 individuals. Of the 125 individuals, 111 were recorded to have been given the expanded plan option, resulting in a sample of 111 for the expanded plan. Figure 1 describes the sample selection.

The sociodemographic variables originally extracted from recipient charts included age, gender, income, and household composition. Of these, only age and income were significantly related to other variables of interest (e.g., ADL score) and were included in the analysis. While important to study, gender was not significantly related to any other variable of interest and, hence, was not included. The typical average recipient was 74.8 years old, with a monthly income of 1436 USD, an ADL score of 3.9, an IADL score of 10.1, and an NRA score of 8.3. It is worth noting that ADL and IADL scores had a significant positive correlation with age, while NRA scores did not (details included in Appendix A). The average number of medical conditions was 2.8, as shown in Table 1. Table 1 also shows the mean values for the basic characteristics of those in the standard plan and those who were selected to opt in to the expanded meal plan. Participants that opted in to the expanded meal plan did not have significantly different functional status levels, nutritional status levels, or medical conditions compared to the participants that were in the standard meal plan.

The nutritional content of hot and chilled standard meals (before recipients opted into the frozen expanded meal plan) was not correlated with age but was correlated with income level, as shown in Table 2. While the measure of income was self-reported with several missing values, leaving us with a sample of only 68 recipients for this table, nutritional values were consistently and significantly higher for those reporting lower incomes, including for Na and CHO; hence, they are highlighted here.

All 125 of the recipients in this sample in the HDM program were assigned a standard hot or chilled meal upon enrolling in HDMs. Table 3 presents the nutritional content of the standard meals at recipients’ start of HDM program enrollment. Of the 125 recipients in this sample, 111 opted in to the expanded meal plan. Table 3 also presents the *default* nutritional values of meals offered through the expanded meal plan compared to the standard hot/chilled meals.

This study’s primary focus was on understanding the nutritional choices that were being made by the recipients of HDMs when given the choice of selecting their own meals from the expanded meal plan. Of the 111 recipients of HDMs who opted in to the expanded meal plan, 40 opted to not change their meals and remained on the default frozen meal choices, while the remaining 71 selected their own preferred meals under the expanded plan. In Table 4, the nutrition value of meals is contrasted between those who opted to leave their meals unchanged (stayed on default frozen meals) compared to those who opted to select their own preferred meals from the expanded meal plan menu. As can be seen from the last two columns of Table 4, values of all nutrients were significantly lower for those who selected their own preferred meals from the expanded menu. The percentage reduction in nutrient levels was highest for potassium, cholesterol, and calories but, perhaps most noteworthy, is the across-the-board reduction, including in protein levels. Table 5 compares the difference in medical conditions between those who opted to leave their frozen meals unchanged (default) and those who selected their own meals from the expanded meal plan. Those who selected their own meals were significantly more likely than those who left their meal unchanged to have cardiac, renal, and thyroid disorders.

## 4. Discussion

This study examined the extent to which HDM recipients were able to select nutritious meals from an expanded HDM meal plan and examined the health characteristics of HDM recipients with the highest need for RDN services. Although expanded meal options may provide HDM recipients with greater autonomy over their meal choices, the present findings, especially in Table 4 and Table 5, indicate that nutrition support and guidance are needed to ensure that recipients are selecting healthy and appropriate meal options, particularly as they pertain to recipients with complex health histories. Among this study’s sample, HDM recipients who selected their own meals opted to receive meals that, on average, were lower in protein and calories compared to default meals offered in the expanded menu plan. It has been well established that protein–energy malnutrition is associated with health decline among older adults [27,28,29], and findings draw attention to the urgent need to assist HDM recipients in their selection of meals to sufficiently meet nutritional intake recommendations. Amongst those with the highest need for RDN services were HDM recipients with a history of cardiovascular disease and kidney disorders, as recipients with these conditions were more likely to opt in to the expanded meal plan. Cardiovascular disease and kidney disorders are also conditions heavily affected by dietary behaviors and nutritional intake [30,31], further suggesting that recipients with these conditions should receive skilled RDN services when selecting meals to match their nutritional needs. In addition, as seen in Table 2, the intake of nutrients such as Na and CHO was significantly higher for recipients with lower levels of income, pointing to the need to address less healthy eating habits among lower income segments of the population in order to foster healthy aging.

The present findings have broad implications for HDM programs nationwide. Firstly, although daily, hot and chilled HDMs provide recipients with the added benefit of socially interacting with their deliverers [32], weekly and biweekly models for delivering frozen HDMs are becoming more popular across the country [33]. With the increase in provision of frozen HDMs, RDNs should provide nutrition education and counseling to ensure that recipients select meals that meet their daily, nutritional needs and align with dietetic recommendations. Secondly, while this study’s findings are drawn from a single HDM agency, the health characteristics of the study’s sample are representative of HDM recipients across the United States. Common health conditions reported by larger samples of HDM recipients include hypertension (90%) and heart failure (47%) [34], as well as diabetes [6]—a condition that is associated with kidney disease and kidney failure [35]. The present study identified that recipients living with kidney disorders and cardiac disorders were more likely to opt in to selecting their own meals from the expanded meal plan menu, suggesting that recipients with these health histories should be targeted for RDN guidance when selecting meal options. Lastly, recent amendments to the Older Americans Act have encouraged HDM providers to implement malnutrition screenings in an effort to promote health and prevent the worsening of diet-related health diagnoses [36,37]. As malnutrition screenings become implemented more frequently among HDM recipients, follow-up RDN services will be critical for helping recipients address or reverse identified factors and behaviors that can lead to malnutrition and, therefore, promote healthy aging.

As older adults continue to express a desire to age in their own homes and communities, rather than residential care facilities [38], it is necessary that HDM providers be well equipped to manage older adults’ complex and ever-changing health and nutritional needs to optimize wellbeing. Recipients of HDMs often present with multiple health comorbidities and experience food insecurity at a level greater than the general older adult population [39,40]. Food-insecure older adults are also more likely to have poor diet quality [39], further validating the importance of RDNs to intervene with nutritional guidance for this vulnerable group. For instance, among older adults with type 2 diabetes, recent evidence indicated that RDNs play a critical role in promoting older adults’ healthy behaviors, particularly in the areas of glycemic control, weight management, and cardiovascular outcomes [34]. RDN services can also provide value to the growing number of HDM recipients living with dementia. Nearly 30% of HDM recipients are living with dementia [41] and have difficulty completing basic self-care (e.g., eating) and home management tasks (e.g., meal preparation) independently [42]. Relatedly, older adults who presented with high energy intake in the forms of carbohydrates, fat, or sugar were more likely to have severe cognitive impairments compared to older adults with lower energy intake [43]. With the anticipated growth community-dwelling older adults living with the dementia [44], RDNs may be well positioned to provide nutritional guidance to help older adults consume well-balanced meals, potentially attenuating the rate of cognitive decline.

RDN services hold great promise not only for improving the dietary behaviors of older adult populations, but also for addressing nutrition-related health disparities for older adults who may have difficulty accessing specialized healthcare services [45,46]. Broadly, healthcare disparities include variability in service access, service quality, and health outcomes across different racial, ethnic, gender, and income groups [47]. As an example, a report by AHRQ found that poor and low-income households in the US received lower-quality healthcare services than the services provided to high-income households [48]. Furthermore, diet-related diseases, such as cardiovascular disease and diabetes, are significantly more likely to affect people of color compared to white individuals [49,50]. Arguably, RDN services could be leveraged to provide low-income and communities of color with skilled dietetic guidance to increase healthy lifestyle behaviors and mitigate the risk of diet-related disease. The present study’s findings suggest that income may be a predictor of insufficient nutritional intake, further validating the importance of RDN services to address the nutritional needs of low-income older adult groups. In light of the findings from the present study, staff and administrators at the study’s partner agency developed a novel referral program to connect HDM recipients to RDN services, particularly for older adults opting to receive the agency’s expanded meal plan. This referral program is described below, and preliminary data are presented that speak to the program’s reach and opportunities for program expansion.

*RDN referral program.* After enrolling in HDMs at the partner agency, older adults completed an initial evaluation with HDM assessment staff member who obtained information pertaining to each older adult’s socio-demographics, income, nutritional status, functional status, and health history. During this initial evaluation period, HDM staff members were encouraged to inform new HDM enrollees that, if interested, enrollees could be referred to one of the agency’s RDNs who could provide basic nutrition education, evaluate the need for specialized meals, and assist in the selection of specialized meal choices. If interested in RDN services, HDM assessment staff notified the agency’s internal case manager, who coordinated referrals to an RDN (see Figure 2 for RDN referral program process). Upon receiving each referral, RDNs contacted HDM recipients via phone to provide nutrition education service calls at no cost to the recipient. During each call, RDNs provided basic nutrition education, reviewed the nutritional content of meals, identified nutrition education materials that could be sent to the older adult in the mail, and provided guidance to recipients in their selection of meals from the expanded meal plan. Although the nutritional needs of recipients were quite varied, the typical length of RDN phone calls with recipients ranged from 15–20 min in duration. As an illustrative example, one recipient was unable to continue with his chemotherapy treatment schedule due to extreme weight loss. Through the RDN referral program, an agency RDN assisted the recipient in selecting meals that were high in protein and calories to promote healthy weight gain.

From 1 September 2021 to 10 December 2021, RDNs provided 129 phone calls to HDM recipients in need of nutrition education and assistance with meal selection from the expanded meal plan menu. Despite the value of these phone calls, such RDN services provided to HDM recipients are not currently billable or reimbursable. Given that access to higher-quality, nutritious meals is associated with improved health outcomes [51,52], the RDN services provided through this referral program have great potential to minimize the risk of health decline among the vulnerable HDM population. This point is particularly important as healthcare payers (e.g., Medicare) shift to value-based payment models, where providers, such as clinicians and hospitals, are reimbursed according to their patients’ health outcomes and face financial penalties if particular quality metrics (e.g., hospital readmission rates) are not met [53,54]. As such, healthcare providers have financial incentives to improve the health outcomes of their patient populations, yet community-based organizations, such as HDM agencies, who provide services and supports that contribute to these improved outcomes, do *not* receive the same financial benefits as their clinician and hospital counterparts. In light of value-based payments, one can argue that HDMs and associated RDN services play a key role in optimizing the health outcomes of community-dwelling older adults, particularly those older adults with conditions (e.g., diabetes, cardiovascular disease, and dementia) that place them at high risk of health decline, and such services should be billable and reimbursable given their well-established benefit for older adults and the entire US healthcare system [4,55]. Going forward, plans to evaluate the impact of RDN services on the nutritional outcomes of the agency’s HDM recipients are ongoing. Data are being collected that represent the frailty levels of HDM recipients, specifically, metrics that speak to recipients’ unintentional weight loss, nutrient intake, and loss of appetite [56].

## 5. Limitations

Although this study makes valuable contributions to the HDM literature, there are some limitations worth noting. Firstly, data drawn from the agency’s electronic health record represent data from recipients who were actively receiving meals during calendar year 2021 and may not be representative of earlier recipients participating in the expanded meal plans. However, while data were drawn from a convenience sample, there is growing evidence to suggest that the partner agency serves a geographic region whose demographic profile is nearly identical to that of country more broadly [57,58]. Secondly, data on fiber, iron, and carbohydrate levels were not collected or calculated for analysis in the present study, but efforts are underway to develop a more robust understanding of the nutritional quality of meals provided to older adults in both the standard HDM program and the expanded meal plan. Thirdly, data drawn from recipients’ electronic health records were obtained from recipients via self-report rather than official medical records or claims data; however, this method of data collection is common practice across HDM agencies nationwide. Fourthly, as indicated in Section 3, complete data on key variables such as income were missing for many individuals, restricting the ability to conduct a strong statistical analysis. Lastly, reductions in sodium and cholesterol, as identified by this study’s analysis of meals provided to recipients in the expanded meal plan, may be perceived as nutritionally beneficial; however, given the well-established importance of protein and energy intake among older adults, it is perceived that HDM recipients should still receive skilled guidance when selecting meals from any expanded plan, especially if recipients have complex diet-related health conditions.

## 6. Conclusions

As the population continues to age at a rapid pace, it is necessary for vulnerable older adults to engage in behaviors that meet their nutritional and dietary needs. Expanded menu plans offered by HDM agencies may provide older adults with greater autonomy over their meal selections—potentially leading to increased food intake—but HDM recipients with multiple chronic conditions need skilled guidance when selecting meal choices that align with their health complexities. RDNs possess the knowledge and expertise to help older adults meet their nutritional needs and can promote healthy eating and dietary behaviors. As HDM agencies nationwide look for strategies to connect their recipients with RDN services, agencies are encouraged to explore models similar to the described RDN referral program with the goal of optimizing older adult nutrition and health-related outcomes.

## Figures and Tables

**Figure 1 nutrients-14-00944-f001:**
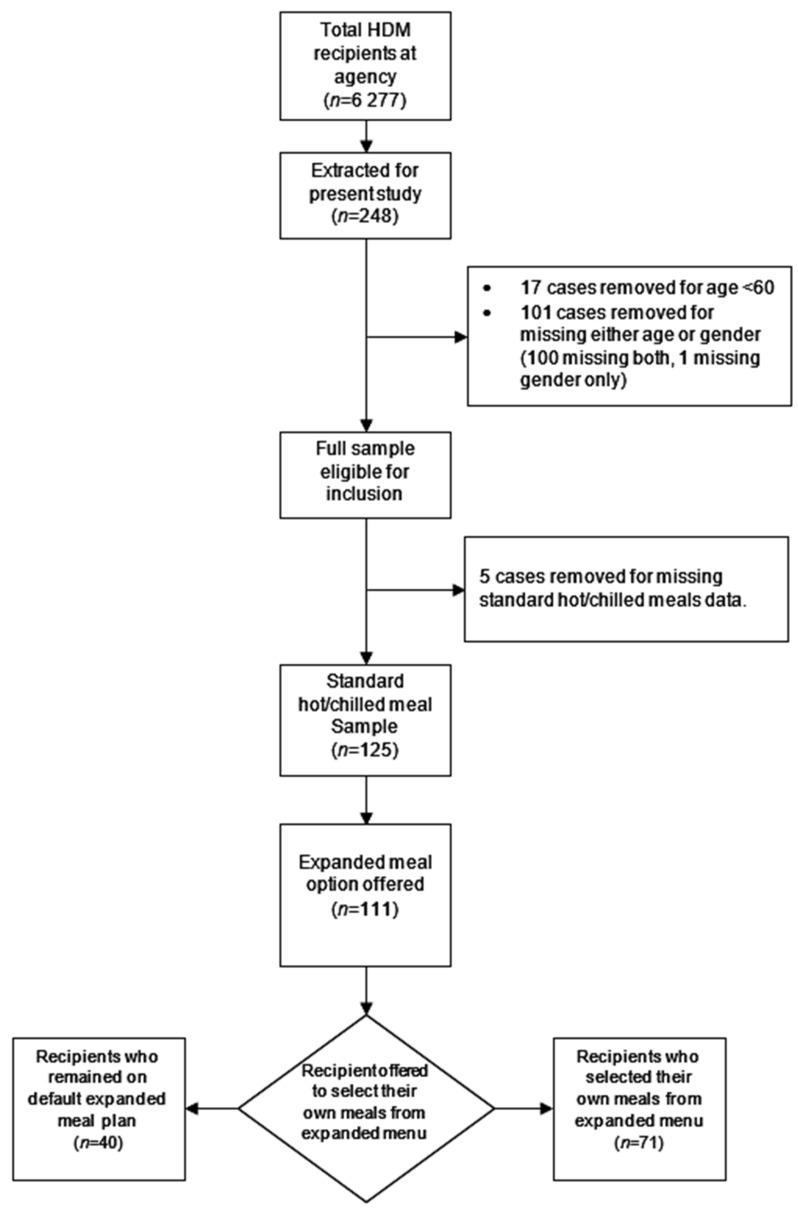
Sample data flowchart.

**Figure 2 nutrients-14-00944-f002:**
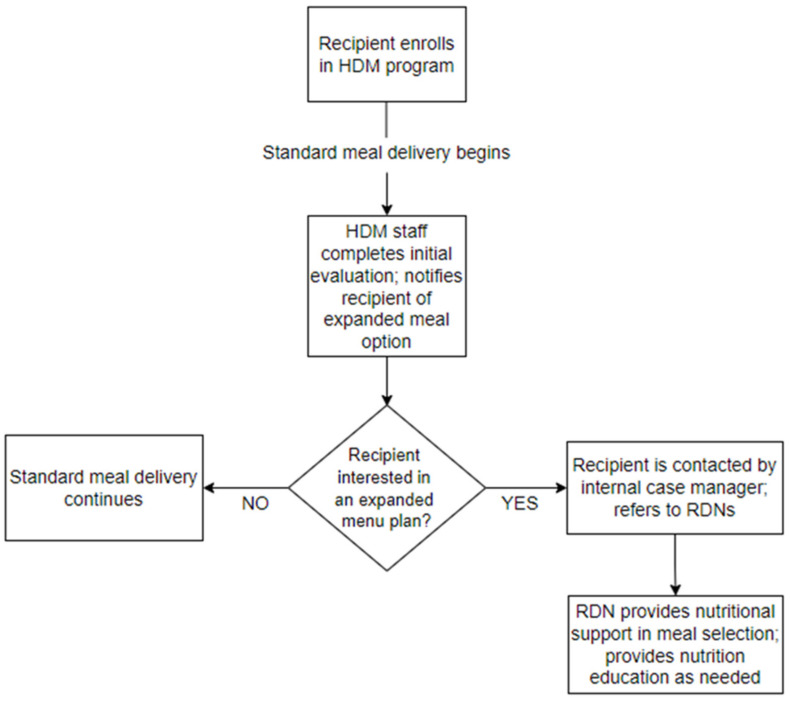
Registered dietician referral program: process chart.

**Table 1 nutrients-14-00944-t001:** Baseline demographic and medical characteristics.

	Full Sample		Standard Plan		Expanded Plan	
Characteristics	*n*	Mean (SD, Range)	*n*	Mean (SD, Range)	*n*	Mean (SD, Range)
Age (years)	130	74.8 (10.7, 41)	125	74.7 (10.7, 41)	111	74.7 (10.8, 41)
Monthly income (USD)	68	1436.3 (1043.7, 7500)	66	1437.6 (1054.2, 7500)	59	1469.5 (1107.4, 7500)
Activities of Daily Living (ADL) score	95	3.9 (4.6, 16)	91	3.7 (4.4, 16)	82	3.7 (4.4, 16)
Instrumental Activities of Daily Living (IADL) score	95	10.1 (7.2, 23)	91	10 (7.2, 23)	82	10 (7, 23)
Nutrition risk assessment (NRA) score	95	8.3 (4, 21)	91	8.2 (4, 21)	82	8.3 (4, 21)
Number of medical conditions	130	2.8 (1.6, 7)	125	2.9 (1.6, 7)	111	2.9 (1.6, 7)

**Table 2 nutrients-14-00944-t002:** Nutrition content of meals by monthly income, prior to expanded meal plan selection (*n* = 68).

Nutrients	Monthly Income (USD)			*p*-Value *
	<2500	2500–5000	>5000	
Protein (g)	29	29	19	<0.001
Sodium (Na) (mg)	811	803	574	<0.001
Fat (g)	18	18	12	<0.001
Cholesterol (CHO) (mg)	68	68	42	<0.001
Calories (kcal)	555	554	360	<0.001
Potassium (K) (mg)	1019	1024	649	<0.001

* *p*-Value on Pearson’s correlation coefficients.

**Table 3 nutrients-14-00944-t003:** Nutritional content of standard hot and chilled meals compared to default meals in the expanded meal plan (*n* = 125).

Nutrients	Standard Hot/Chilled		Default Expanded		Default—Standard	
	*n*	Mean (SD, Range)	*n*	Mean (SD, Range)	*n*	Mean (SD, Range)
Protein (g)	125	28.9 (1.2, 11.1)	111	25.5 (3.9, 19.2)	111	−3.4 (3.8, 17.1)
Sodium (Na) (mg)	125	801.5 (54.6, 487.8)	111	734.3 (144.7, 852.4)	111	−67.1 (138.8, 852.4)
Fat (g)	125	17.6 (1.3, 11.5)	111	15.7 (3.4, 20.1)	111	−1.9 (3.4, 20.1)
Cholesterol (CHO) (mg)	125	67.2 (5.6, 42.40)	111	53.3 (13.5, 50.8)	111	−15.9 (13.8, 57.7)
Calories (kcal)	125	548.5 (35.3, 272.4)	111	461.4 (86.5, 295.6)	111	−86.4 (88.7, 303.6)
Potassium (K) (mg)	125	1009.1 (77.3, 603.4)	111	776.2 (212.4, 797.3)	111	−231 (216.5, 797.3)

**Table 4 nutrients-14-00944-t004:** Nutritional content of default expanded meals compared to expanded meals selected by recipients (*n* = 111).

	Recipients Who Remained on Default Expanded Meals		Recipients Who Selected Their Own Expanded Meals		Difference	
	*n*	Mean (SD, Range)	*n*	Mean (SD, Range)	Mean (SD, Range)	*p*-Value
Protein (g)	40	28.5 (2.1, 11.1)	71	23.8 (3.6, 19.2)	−5.3 (3.6, 17.1)	<0.001
Sodium (Na) mg	40	789.1 (86.5, 487.8)	71	703.4 (161.4, 852.4)	−104.9 (162, 852.4)	<0.001
Fat g	40	17.2 (1.8, 8.6)	71	14.9 (3.8, 20.1)	−2.9 (3.9, 20.1)	<0.001
Cholesterol (CHO) (mg)	40	64.8 (9.2, 42.4)	71	46.9 (11, 50.8)	−21.5 (11.3, 57.7)	<0.001
Calories (kcal)	40	532.7 (57.4, 232.5)	71	421.2 (73.4, 295.6)	−135 (75.6, 303.6)	<0.001
Potassium (K) (mg)	40	975.6 (131.3, 603.4)	71	663.8 (160.5, 797.3)	−361.2 (161.1, 797.3)	<0.001

**Table 5 nutrients-14-00944-t005:** Medical conditions associated with expanded meal plan (*n* = 111).

	Recipients Who Remained on Default Expanded Meals			Recipients Who Selected Their Own Expanded Meals			
	*n*	Proportion	Std. Dev.	*n*	Proportion	Std. Dev.	*p*-Value
Cardiac disorder	40	0.1	0.304	71	0.282	0.453	0.0132
Renal disorder	40	0.05	0.221	71	0.197	0.401	0.0141
Thyroid disorder	40	0	0	71	0.085	0.28	0.0132

## Data Availability

Due to ethical restrictions and client confidentiality, data cannot be made publicly available. However, data from the RDN referral program are available upon request, for researchers who meet the criteria for access to confidential data. Please direct your requests to the corresponding author (Lisa A. Juckett).

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
