# Peer review of "Implementing a Community-Based Initiative to Improve Nutritional Intake among Home-Delivered Meal Recipients"

_nutrients, 2022, doi:10.3390/nu14050944_

Round 1
Reviewer 1 Report
- Review the discussion section. I suggest to give more space to the scientific findings as well as the findings' external validity . If it is valuable that recomendations be included in a socially relevant paper, extensive details on the RDN referral program would be appropiatte in a paper showing the results of the program testing.
- 2. Review paragraph in page 4, lines 168-173, it seems that a line is repeated.
Author Response
Thank you for the opportunity to revise our manuscript, “Implementing a community-based initiative to improve nutritional intake among home-delivered meal recipients,” for this special issue in Nutrients. We sincerely appreciate your comments which have substantially improved the quality of this manuscript. Below, we describe how we addressed each comment and provide original and revised line numbers [in brackets] indicating where our revisions were inserted.
Reviewer 1:
- Review the discussion section. I suggest to give more space to the scientific findings as well as the findings' external validity. If it is valuable that recommendations be included in a socially relevant paper, extensive details on the RDN referral program would be appropriate in a paper showing the results of the program testing.
- We agree with this comment. In the discussion section, we have expanded on our scientific findings with stronger claims for how our findings can be generalized to the wider HDM population and other HDM agencies [see lines 222-241].
- Review paragraph in page 4, lines 168-173, it seems that a line is repeated.
- We have reviewed this section and verified that no content/phrasing has been repeated.
Reviewer 2 Report
The manuscript aimed to answer 1) To what extent are HDM recipients able to select nutritious meal options from an expanded HDM meal plan? and 2) What are the health characteristics of HDM recipients in highest need of RDN services? And present a referral model, currently being implemented by one HDM agency, to streamline RDN services to HDM recipients, a subgroup of older adults at high risk for malnutrition and associated health disparities. The theme is interesting, but it is necessary for some reviews that I highlight below.
- Line 59 – Why not?
- Line 72 – What does OAA mean?
- Lines 97-102 – Include the exclusion criteria. Was your sample a convenience sample?
- Lines 120-121 – How was the adaptation. Was the new version tested and validated? Were the items' numbers and scores the same as the original ones?
- Lines 133 – Why did you not evaluate carbohydrates, fiber, and iron? They are important for the elderly and should be included.
- Lines 135 and 136 – mean and standard deviation.
- Line 155 – Include the total of the individuals who met the inclusion criteria. Include the % of missings and justify your sample size.
- Lines 157 – 160 – It should be included in the method section. It is not result.
- Lines 163-164 – include data as a supplementary file.
- Lines 231-246 – introduction section and part of (up to line 277) the discussion (as data collection) can be placed in the method section.
- Figure 1 – the title should be placed below the figure. The title should not contain acronyms.
Thank you for the opportunity to review this manuscript!
Author Response
Thank you for the opportunity to revise our manuscript, “Implementing a community-based initiative to improve nutritional intake among home-delivered meal recipients,” for this special issue in Nutrients. We sincerely appreciate your comments which have substantially improved the quality of this manuscript. Below, we describe how we addressed each comment and provide original and revised line numbers [in brackets] indicating where our revisions were inserted.
Reviewer 2:
- Line 59 – Why not?
- This opening sentence has been rephrased to add clarity to the value of RDN services [line 60].
- Line 72 – What does OAA mean?
- We have spelled out this acronym: Older Americans Act [line 73]
- Lines 97-102 – Include the exclusion criteria. Was your sample a convenience sample?
- We agree that greater clarity is needed here. We have now added language to explain our inclusion/exclusion criteria that were applied to our original sample [lines 102-108].
- Lines 120-121 – How was the adaptation. Was the new version tested and validated? Were the items' numbers and scores the same as the original ones?
- We recognize that this is a limitation of these measures; however, these measures are mandated for use by the State Unit on Aging overseeing HDM programs in our state. Rectifying this issue is a matter of policy change that we hope our future work can influence state’s selection of standardized assessments implemented by HDM agencies.
- Lines 133 – Why did you not evaluate carbohydrates, fiber, and iron? They are important for the elderly and should be included.
- This is a very legitimate point that has been addressed in our limitations section as the macronutrients were evaluated were those that were made available to use by the meal vendor [lines 340-343]
- Lines 135 and 136 – mean and standard deviation.
- These values have been updated in Supplemental Table 1.
- Line 155 – Include the total of the individuals who met the inclusion criteria.
- Thank you. We have updated these numbers in our methods section [lines 102-108].
- Lines 157 – 160 – It should be included in the method section. It is not result.
- Similar to the comment above, we have clarified this language [lines 102-108].
- Lines 163-164 – include data as a supplementary file.
- Thank you for this suggestion—we have now added supplementary tables 1-4.
- Lines 231-246 – introduction section and part of (up to line 277) the discussion (as data collection) can be placed in the method section.
- Thank you for this suggestion. We have inserted additional background information into the introduction section so that there is greater synergy between the introduction and discussion sections [lines 88-92].
- Figure 1 – the title should be placed below the figure. The title should not contain acronyms.
- Thank you. Figure 1 has been updated [starting at line 289].
Round 2
Reviewer 2 Report
The authors well improved the manuscript. However, some information is still lacking which impairs the quality of the study. It is not possible to judge the study method lacking information that I mentioned in the 1st review. Also, the changes should be done using word tracking changes or similar to make the reviewers' evaluation easier.
1) Despite not being inserted in the table, you should estimate the carbohydrate content using data from fat, protein and calories to discuss it.
2) It is not clear if you use a convenience sample. Is your sample representative? Did you calculate the minimum sample?
3) It is not clear if the authors tested and validated the version adapted.
4) Fiber and iron should be mentioned in the limitations if you don't have this information. However, if you do have the information, please insert it in the manuscript.
5) The Tables' acronyms must be explained below each table. Tables must be "self-explanatory"
6) I did not understand the second "table 5" (line 195). It is not "nutritional content of meals". Should it be table 6 with another title?
Author Response
Dear Editor and Reviewer 2:
Our sincere thanks to Reviewer 2 for raising some important points of clarification that have helped to improve the manuscript. Below we have provided an itemized response for the reviewer’s questions (bullet points). We have also now provided the manuscript with the changes tracked throughout for ease of evaluation, as suggested by the reviewer.
1) Despite not being inserted in the table, you should estimate the carbohydrate content using data from fat, protein and calories to discuss it.
- We thank the reviewer for this important point. We have added text (line 151) to clarify this limitation and also acknowledged future efforts being deployed to collect more robust data to understand the nutritional value of HDMs (line 343)
2) It is not clear if you use a convenience sample. Is your sample representative? Did you calculate the minimum sample?
- Thanks to the reviewer for this question. The sample was a selection of data from clients who opted to receive HDM in one region and hence is not representative. There is, however, some evidence that this region is an excellent representation for the nation. We did not calculate a minimum sample but in order to sufficiently answer the research questions kept all the recipients with complete health and demographics documented (lines 103-109) This is a limitation and has now been mentioned in the study limitations (Beginning with line 346).
3) It is not clear if the authors tested and validated the version adapted.
- The ADL and IADL measures used did not undergo psychometric evaluation. They are administered to all recipients who enrolled in HDM and are mandated for use by the region’s state unit on aging. We have clarified this in the manuscript (Beginning with line 128)
4) Fiber and iron should be mentioned in the limitations if you don't have this information. However, if you do have the information, please insert it in the manuscript.
- We agree with the reviewer that fiber and iron are important nutritional elements. Unfortunately, we do not have the fiber and iron content information in our data and have now mentioned it in the study limitations (Beginning with line 343).
5) The Tables' acronyms must be explained below each table. Tables must be "self-explanatory.
- We thank the reviewer for this comment and have now included the full description of the acronym in all the tables for clarity.
6) I did not understand the second "table 5" (line 195). It is not "nutritional content of meals". Should it be table 6 with another title?
- Our apologies for the error and thanks to the reviewer for pointing this out. The numbering and title have now been corrected and is now Table 6 “Medical Conditions and Choice of Expanded Meal Plan” (line 215).
Thank you again for the thoughtful and helpful comments!